# Pregnancy exposure registries for drugs and vaccines in low-income and middle-income countries: scoping review protocol

Rahmeh AbuShweimeh,[1] Sophie Knudson,[2] Sonia Chaabane,[3] Shanthi Narayan Pal,[3] Becky Skidmore,[4] Andy Stergachis [ID] ,[1,5] Niranjan Bhat[2]

¹School of Public Health, University of Washington, Seattle, Washington, USA
²Center for Vaccine Innovation and Access, PATH, Seattle, Washington, USA
³Regulation and Prequalification, World Health Organization, Geneve, Switzerland
⁴Independent Information Specialist, Ottawa, Ontario, Canada
⁵School of Pharmacy, University of Washington, Seattle, Washington, USA

**Correspondence to**
Professor Andy Stergachis;
stergach@uw.edu

## ABSTRACT

**Introduction** Data regarding the safety of drugs and vaccines in pregnant women are typically unavailable before licensure. Pregnancy exposure registries (PERs) are an important source of postmarketing safety information. PERs in low-income and middle-income countries (LMICs) are uncommon but can provide valuable safety data regarding their distinct contexts and will become more relevant as the introduction and use of new drugs and vaccines in pregnancy increase worldwide. Strategies to support PERs in LMICs must be based on a better understanding of their current status. We developed a scoping review protocol to assess the landscape of PERs that operate in LMICs and characterise their strengths and challenges.

**Methods and analysis** This scoping review protocol follows the Joanna Briggs Institute manual for scoping reviews. The search strategy will be reported using the Preferred Reporting Items for Systematic Reviews and Meta-Analyses extension for Scoping Reviews Checklist. We will search PubMed, Embase, CINAHL and WHO's Global Index Medicus, as well as the reference lists of retrieved full-text records, for articles published between 2000 and 2022 that describe PERs or other resources that systematically record exposures to medical products during pregnancy and maternal and infant outcomes in LMICs. Title and abstracts will be screened by two authors and data extracted using a standardised form. We will undertake a grey literature search using Google Scholar and targeted websites. We will distribute an online survey to selected experts and conduct semistructured interviews with key informants. Identified PERs will be summarised in tables and analysed.

**Ethics and dissemination** Ethical approval is not required for this activity, as it was determined not to involve human subjects research. Findings will be submitted to an open access peer-reviewed journal and may be presented at conferences, with underlying data and other materials made publicly available.

## INTRODUCTION

Newly introduced vaccines and drugs hold the promise of reducing morbidity and mortality among pregnant women and infants living in low-income and middle-income countries (LMICs). However, since pregnant women are actively excluded from

---

## STRENGTHS AND LIMITATIONS OF THIS STUDY

⇒ This scoping review protocol identifies the landscape of pregnancy exposure registries that operate in low-income and middle-income countries and characterises their strengths and challenges for use in assessing the safety of new vaccines introductions.
⇒ The methodology is guided by the Joanna Briggs Institute Scoping Review Manual and the reporting will be informed by the Preferred Reporting Items for Systematic Reviews and Meta-Analyses checklist and an expert technical working group.
⇒ To augment limitations associated with searches of bibliographic databases and grey literature, experts and key informants will be queried to identify additional pregnancy exposure registries and systems that record exposure to medical products during pregnancy and maternal and perinatal outcomes in low-income and middle-income countries.

---

most preregistration clinical trials, safety information for this group is rarely available at the time of a medical product's licensure or approval.[1 2] Consequently, the safety of drugs and vaccines administered during pregnancy must be evaluated throughout their life cycle, including through active surveillance approaches during the postlicensure or postauthorisation phase. A commonly used method to systematically assess postapproval safety of drugs and vaccines in pregnant women and their offspring is through the use of a pregnancy exposure registry (PER). A PER is an observational study that systematically collects health information on exposure to medical products such as drugs and vaccines during pregnancy.

PERs, particularly in high-income countries (HICs), are used throughout the postmarketing phase to monitor the safety of drugs and vaccines used during pregnancy.[3 4] PERs have been used infrequently in LMICs, where there are unique challenges with respect to the knowledge of background rates for obstetric and neonatal outcomes, access to the interventions under

evaluation, and the availability of data collection resources and infrastructure. This latter point underscores the limitations in capturing information on the use of vaccines and other medicines in pregnancy, as well as information on the occurrence of obstetric/perinatal complications and other outcomes, and the capacity to link these data sources together. Examples of PERs that operate in LMICs include those established for drugs or vaccines of particular relevance to their populations, such as those used to treat malaria or HIV, or to prevent COVID-19.[5–7] Maternal immunisation (MI) is an effective method to protect women during pregnancy and their newborns, and their use in LMICs is growing.[8] MI is anticipated to increase further in the coming years as a result of the continued adoption of COVID-19 vaccines and the introduction of promising new MI-specific vaccines, such as those for respiratory syncytial virus and group B *Streptococcus.* Prelicensure safety data on these vaccines will largely come from HIC settings, highlighting the importance of monitoring their postmarketing safety in LMICs.[9 10]

Most research assessing the global status of drug and vaccine safety monitoring in pregnancy has focused on HICs.[3] However, one recent study assessed existing maternal, newborn and child health data collection systems in LMICs that could be used to monitor drug or vaccine safety.[11] Another study assessed the feasibility of use of Global Alignment of Immunisation Safety Assessment in pregnancy (GAIA) case definitions for neonatal outcomes and maternal vaccination in LMICs.[12] In contrast to these broader surveillance systems, PERs focus on active data collection specifically related to medical product exposures during pregnancy and pregnancy safety outcomes, and may be conducted by private as well as public agencies.[13] An improved understanding of PERs in LMICs can better inform how future public health efforts, such as new vaccine introductions and treatment programmes, can supported maternal populations. To address this need, we aim to conduct a scoping review to identify and describe PERs and other similar resources that operate in LMICs. Here, we report the methodology for our planned scoping review.

## Study objective

This scoping review protocol aims to identify PERs, databases and other routinely collected health data that systematically record exposures to medical products during pregnancy and maternal and infant outcomes in LMICs. The scoping review will consist of a systematic search of the scientific and grey literature, supplemented by an online survey and interviews with selected key informants, as needed.

## METHOD
### Protocol design

This scoping review protocol follows the Joanna Briggs Institute manual for scoping reviews, and the search strategy will be reported using the Preferred Reporting Items for Systematic reviews and Meta-Analyses extension

for Scoping Reviews (PRISMA) Checklist.[14–16] This protocol has been registered with the Open Science Framework (DOI: https://doi.org/10.17605/OSF.IO/FU5AT).[17] The scoping review start date and estimated end date are 1 July 2022 and 30 June 2023, respectively.

### Review questions

Following multiple consultations with key stakeholders, three primary review questions were selected:
1. What pregnancy exposure cohorts, databases and registries exist in LMICs?
2. What types of data, processes and tools are included in these databases and registries?
3. What are the strengths and weaknesses of the identified databases and registries?

In addition, two secondary questions were identified:
1. Can the PERs that have been identified in LMICs be used or adapted to monitor additional new vaccines or drugs that may be introduced for pregnant women?
2. What is the potential for data harmonisation and/or combining of data across databases and registries?

Based on these objectives, the following eligibility criteria for selection were developed:

### Inclusion criteria

1. Publications and documents published or produced from January 2000 to the present, to ensure that identified registries possess features that are more relevant to current scientific and technological conditions; online sources will be accessed.
2. Populations studied are located entirely or at least partially in LMICs.[18]
3. Reference to prospective and retrospective electronic or combined paper-electronic data collection systems, including demographic national registers in LMICs.
4. Reference to prospective and retrospective cohort studies, with no restrictions regarding age range other than women of childbearing age (ie, 15–49 years of age) or underlying conditions other than pregnancy.
5. Reference to systems that collect data on exposure to one or more drugs or vaccines during pregnancy.
6. Reference to systems that collect data on pregnancy outcomes, including delivery, post partum and neonatal outcomes (may include an extended time frame to include birth defects detected later).

### Exclusion criteria

1. Editorials, opinion pieces, promotional literature.
2. Guidelines or guidance documents.
3. Reference to non-allopathic (eg, traditional, homeopathic or naturopathic) interventions.

### Search strategy and information sources

Using an iterative process and in consultation with the review team, an information specialist will develop a strategy in PubMed incorporating controlled vocabulary/Medical Subject Headings (eg, "Pregnancy", "Datasets as Topic", "Product Surveillance, Postmarketing") and free text (eg, prenatal, registries, pharmacovigilance)

(see online supplemental appendix 1). We will apply an LMIC filter to focus results to the geographical regions of interest. Another information specialist will peer review the strategy using the PRESS Checklist.[6] Any necessary edits will be made before finalising in PubMed and subsequently translating the search to Embase, CINAHL and WHO's Global Index Medicus. We will also search the reference lists of potentially relevant records and articles to ensure that our search results are as comprehensive as possible.

In addition, we will undertake a grey literature search, including a Google Scholar search and review of relevant websites, such as industry and professional organisations, associations and alliances (eg, Developing Countries Vaccine Manufactures Network, Society for Maternal-Fetal Medicine, American College of Obstetricians and Gynecologists, International Society for Pharmacoepidemiology, International Society of Pharmacovigilance); selected Ministries of Health (including regulatory agencies and pharmacovigilance centres) in LMICs; and selected HIC organisations, including US Food and Drug Administration, European Medicines Agency, US National Institutes of Health, WHO, Maternal and Child Survival Programs, Measure Evaluation, UK Teratology Information Service, UK Obstetric Surveillance System, and selected academic and other non-governmental groups. The final search strategy is provided in online supplemental appendix 1.

## STUDY SELECTION

Records retrieved by the search strategy will be downloaded to EndNote V.9.3.3 (Clarivate) for deduplication and then uploaded to review management software (Covidence) to enable independent screening and track disagreements and consensuses among reviewers. Each title and abstract will be screened by two independent reviewer authors to determine eligibility. Each item will be categorised into one of three categories (yes, maybe, no), and disagreements between reviewers will be resolved by a third reviewer. After title and abstract screening, a second round of screening will be conducted for full-text review by two reviewers and a decision will be made for data extraction. An adapted version of the PRISMA flow diagram will be constructed to summarise the number of records screened, assessed for eligibility and included in the review, with reasons for exclusions at each stage.[7]

### Data extraction

A form for data extraction will be used to extract key information regarding the registries from the selected full-text articles and screened grey literature (see online supplemental appendix 2). The form was pilot-tested and refined during the full-text screening stage in order to capture information more efficiently. Key data elements

to be collected from the included articles include the following:

a. Author(s).
b. Year of publication.
c. Country(ies) where the registry is located.
d. Name and aims/purpose of the registry.
e. Years of registry operation.
f. Characteristics of the included population (specifically, the characteristics or eligibility criteria used to be included in the registry).
g. Country representativeness (local/national/regional).
h. Funding source for the registry.
i. Current sample size (proportion in LMIC).
j. Methodology/methods, including database type.
k. Terminology and data system used (eg, MedDRA, International Classification of Diseases (ICD), Brighton Collaboration).
l. Intervention/exposure type (eg, drugs, vaccines), comparator and details of these, including trimester of exposure.
m. Maternal, perinatal and neonatal outcomes including, but not limited to spontaneous abortions, stillbirths, congenital anomalies and details of these. In addition, documenting whether GAIA definitions or other standardised classifications are used.
n. Duration of follow-up.
o. Key findings that relate to the scoping review questions, including strengths, weaknesses, ability to add new interventions, and ability to combine data with other systems.
p. Demographic factors, socioeconomic factors, and lifestyle factors.

Data will be entered into a database that will allow searching and categorizsation according to selected characteristics.

### Informant survey and interviews

In addition to searching bibliographic databases and grey literature, an online survey will be sent to experts and key informants to identify additional PERs, surveillance registries, databases and routinely collected data that record exposure to medical products during pregnancy and maternal and perinatal outcomes in LMICs that may not have been captured or to provide additional detail on those already identified. Information obtained through key informant interviews will be subject to the same inclusion and exclusion criteria as used with the literature. A subset of survey respondents will be identified for semistructured interviews if additional information is needed about the registries. Interviewees will be asked to discuss the strengths and weaknesses of the registry, relevant contextual factors and the usability of these resources in their particular settings. All responses will be recorded in an electronic database for analysis.

### Data analysis

Identified PERs will be summarised in tables organised according to relevant characteristics, including geography,

methodology, types of interventions included, outcomes captured and citations. The selected PERs will be further grouped and evaluated based on the primary and secondary scoping review questions (strengths, weaknesses, ability to add new interventions and ability to combine data with other systems) and other methods of appraisal, and the quality of the existing registries may be discussed as part of the findings. Geographical coverage or other spatial characteristics may be presented using maps.

## Consultation

As part of this activity, a technical working group was established to provide assistance and guidance throughout the course of the review. Members of this group are experts in multiple disciplines, including pharmacovigilance, perinatology and paediatrics, particularly as practised in low-resource settings. In addition to this group, the protocol and results of this scoping review will be reviewed by an Expert Steering Committee on Safety Surveillance in Pregnancy in LMICs, established by WHO. Feedback from both groups will be incorporated to produce a final document.

## Patient and public involvement

There will be no patient or public involvement in this scoping review.

## Dissemination and ethics

Ethical approval is not required for this activity, though it was reviewed by PATH's Research Determination Committee and deemed not to be human subjects research.

Results of this landscape analysis will be submitted to an open-access peer-reviewed journal for publication and may be presented at conferences, while underlying data and other materials will be made publicly available.

**Acknowledgements** The authors would like to thank Kaitryn Campbell for peer review of the PubMed search strategy. We also thank the members of our Technical Working Group: Pierre Buekens, Ushma Mehta, Deshayne Fell, Smaragda Lamprianou, Kate Fay, Ajoke Sobano ter-Meulen, Nancy Salts, and Jessica Fleming. The authors alone are responsible for the views expressed in this article and they do not necessarily represent the views, decisions or policies of the institutions with which they are affiliated.

**Contributors** NB was responsible for conception and design of the review. BS developed the search strategies. SK and RA are conducting the initial review of manuscripts. NB and AS are second reviewers. SC and SNP provided expert review and input into the protocol. RA and AS were responsible for producing the initial draft of the review protocol. All authors provided significant editorial comments on the protocol drafts and read and approved the final manuscript.

**Funding** This work was supported by a grant from the Bill & Melinda Gates Foundation (grant number INV-037810).

**Disclaimer** The funders had no role in the design, analysis, or writing of this manuscript.

**Competing interests** None declared.

**Patient and public involvement** Patients and/or the public were not involved in the design, or conduct, or reporting, or dissemination plans of this research.

**Patient consent for publication** Not applicable.

**Provenance and peer review** Not commissioned; externally peer reviewed.

**ORCID iD**
Andy Stergachis http://orcid.org/0000-0003-0057-6627

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
