## [Reviewer comments · BMJ Open]

ARTICLE DETAILS

TITLE (PROVISIONAL)	Pregnancy Exposure Registries for Drugs and Vaccines in Low- and Middle-Income Countries: Scoping Review Protocol
AUTHORS	AbuShweimeh, Rahmeh; Knudson, Sophie; Chaabane, Sonia; Pal, Shanthi Narayan; Skidmore, Becky; Stergachis, Andy; Bhat, Niranjana

VERSION 1 – REVIEW

REVIEWER	Apte, Aditi KEM Hospital Pune Research Centre
REVIEW RETURNED	02-Feb-2023

GENERAL COMMENTS	1. It is apparent that the scope of the review is limited to drugs and vaccines in pregnancy. The same should be specified in the title of the study. 2. The information on pregnancy registries will be additionally collected from respondents through interviews. Is there any criteria for inclusion of such registries: e.g. minimal dataset or duration, documentation of existence of a pregnancy registry.
--

REVIEWER	Lasky, Tamar US FDA
REVIEW RETURNED	06-Feb-2023

GENERAL COMMENTS	The MS is well written and clear however the scoping review protocol can be strengthened by more closely following the JBI template. In particular, 1) The stated aim is and could be strengthened with specifics/details. It is currently written as “we aim to conduct a scoping review to identify and describe PERs and other similar resources that operate in LMICs” What is included under “other similar resources”? 2) P. 6 Study objective can be strengthened by combining the two sentences so that the aim is clear (i.e., This scoping review aims to identify pregnancy exposure registries, databases and other routinely collected health data that record exposures to medical products during pregnancy and maternal and infant outcomes in LMICs). 3) Section titles do not currently follow the JBI template JBI_Protocol_Template_Scoping_Reviews.docx (live.com) although the authors state, “This protocol follows the Joanna Briggs Institute (JBI) manual for scoping reviews. This is confusing to the reader. For example, while the MS has a sub-section “Study Design” there is no sub-section, “Study Design” in the JBI template.
--

	Other details required in the JBI template are not followed, for example, the Methods section should describe aspects of the Search strategy indicating languages and time period to be included, and the draft data extraction form should be provided in the appendix. 4) Suggest that authors revise in accordance with the JBI Protocol Template for Scoping Reviews.
--	---

VERSION 1 – AUTHOR RESPONSE

Reviewer #1:

1. It is apparent that the scope of the review is limited to drugs and vaccines in pregnancy. The same should be specified in the title of the study.

Response: The title of the manuscript was revised accordingly and now reads: Pregnancy Exposure Registries for Drugs and Vaccines in Low- and Middle-Income Countries: Scoping Review Protocol

2. The information on pregnancy registries will be additionally collected from respondents through interviews. Is there any criteria for inclusion of such registries: e.g. minimal dataset or duration, documentation of existence of a pregnancy registry.

Response: The section on Informant Survey and Interviews was revised accordingly and the first two sentences now reads:

In addition to searching bibliographic databases and grey literature, an online survey will be sent to experts and key informants to identify additional pregnancy exposure registries, surveillance registries, databases and routinely collected data that record exposure to medical products during pregnancy and maternal and perinatal outcomes in LMICs that may not have been captured or to provide additional detail on those already identified. Information obtained through key informant interviews will be subject to the same inclusion and exclusion criteria as used with the literature.

Reviewer 2:

The MS is well written and clear however the scoping review protocol can be strengthened by more closely following the JBI template.

Response: We thank the reviewer for the comment.

The stated aim is and could be strengthened with specifics/details. It is currently written as “we aim to conduct a scoping review to identify and describe PERs and other similar resources that operate in LMICs” What is included under “other similar resources”?

Response: The other similar resources are databases and other routinely collected health data that systematically record exposures to medical products during pregnancy and maternal and infant outcomes in LMICs. This has now been clarified in the section, Study Objective.

P. 6 Study objective can be strengthened by combining the two sentences so that the aim is clear (i.e., This scoping review aims to identify pregnancy exposure registries, databases and other routinely collected health data that record exposures to medical products during pregnancy and maternal and infant outcomes in LMICs).

Response: The Study Objective has been revised accordingly and now reads: This scoping review aims to identify pregnancy exposure registries, databases and other routinely collected health data that systematically record exposures to medical products during pregnancy and maternal and infant outcomes in LMICs.

3) Section titles do not currently follow the JBI template JBI_Protocol_Template_Scoping_Reviews.docx (live.com) although the authors state, "This protocol follows the Joanna Briggs Institute (JBI) manual for scoping reviews. This is confusing to the reader. For example, while the MS has a sub-section "Study Design" there is no sub-section, "Study Design" in the JBI template. Other details required in the JBI template are not followed, for example, the Methods section should describe aspects of the Search strategy indicating languages and time period to be included, and the draft data extraction form should be provided in the appendix.

Response: Under the Methods and subsection, Protocol Design, we have now clarified that the scoping review follows the Joanna Briggs Institute (JBI) manual for scoping reviews. To the extent practical, the revised manuscript has now been revised in accordance with the JBI Protocol Template for Scoping Reviews.

4) Suggest that authors revise in accordance with the JBI Protocol Template for Scoping Reviews.

Response: To the extent practical, the revised manuscript has now been revised in accordance with the JBI Protocol Template for Scoping Reviews.

VERSION 2 – REVIEW

REVIEWER	Apte, Aditi KEM Hospital Pune Research Centre
REVIEW RETURNED	27-Mar-2023

GENERAL COMMENTS	The authors have incorporated all comments.
---

REVIEWER	Lasky, Tamar US FDA
REVIEW RETURNED	29-Mar-2023

GENERAL COMMENTS	The MS is titled "Pregnancy Exposure Registries for Drugs and Vaccines in Low- and Middle-Income Countries: Scoping Review Protocol" however language in MS refers to the MS as a "scoping review" and includes a section called "protocol design". If the
--

	entire MS is a protocol it is NOT a scoping review, itself, but a protocol for conducting a scoping review. The various errors and inconsistencies raise many concerns whether the authors understand what they are doing.
--	--

VERSION 2 – AUTHOR RESPONSE

Reviewer #1:

The authors have incorporated all comments.

Response: We thank the reviewer for this comment.

Reviewer 2:

The MS is titled "Pregnancy Exposure Registries for Drugs and Vaccines in Low- and Middle-Income Countries: Scoping Review Protocol" however language in MS refers to the MS as a "scoping review" and includes a section called "protocol design". If the entire MS is a protocol it is NOT a scoping review, itself, but a protocol for conducting a scoping review. The various errors and inconsistencies raise many concerns whether the authors understand what they are doing.

Response: The revised manuscript clarifies that it is a scoping review protocol. We have addressed any inconsistencies in that regard throughout the revised manuscript.